# Single-Channel FMCW-Radar-Based Multi-Passenger Occupancy Detection Inside Vehicle

**DOI:** 10.3390/e23111472

**Published:** 2021-11-08

**Authors:** Heemang Song, Hyun-Chool Shin

**Affiliations:** Department of Software Convergence, Soongsil University, Seoul 06978, Korea; songhm@soongsil.ac.kr

**Keywords:** FMCW radar, radar signal processing, automotive applications, passenger detection, occupancy detection

## Abstract

In this paper, we provide the results of multi-passenger occupancy detection inside a vehicle obtained using a single-channel frequency-modulated continuous-wave radar. The physiological characteristics of the radar signal are analyzed in a time-frequency spectrum, and features are proposed based on these characteristics for multi-passenger occupancy detection. After clutter removal is applied, the spectral power and Wiener entropy are proposed as features to quantify physiological movements arising from breathing and heartbeat. Using the average means of both the power and Wiener entropy at seats 1 and 2, the feature distributions are expressed, and classification is performed. The multi-passenger occupancy detection performance is evaluated using linear discriminant analysis and maximum likelihood estimation. The results indicate that the proposed power and Wiener entropy are effective features for multi-passenger occupancy detection.

## 1. Introduction

The development of autonomous vehicles has resulted in increasing interest in automobile safety technologies. Typical applications of automobile safety technologies are present in passenger safety systems, such as passenger-side airbags and seat belt reminders. In this technology, whether a passenger is occupying a seat must be determined [1]. In addition, the prevention of children dying of heat stroke inside a vehicle will also be discussed [2,3]. Accordingly, in the European New Car Assessment Programme (Euro NCAP) 2025 roadmap, child presence detection has been added as a safety requirement [4]. In addition, the United States is mandating the use of a rear occupant alert system in cars [5]. Therefore, it is clear that passenger occupancy detection in vehicles has become increasingly important.

Various sensors for detecting passengers have been tested. The most typically used sensor is the pressure sensor [6,7]. However, because passenger detection is based on weight, it is difficult to distinguish between passengers and heavy objects placed on a seat. In addition, because only a single seat can be detected by a single sensor, multiple sensors must be installed. Capacitive sensors [8] that detect the dielectric dispersion effects of biological tissues are affected by high false alarm rates. Camera sensors [9,10] that may be dependent on brightness cannot easily detect passengers in the dark, and consume a high amount of power compared with other sensors. Meanwhile, infrared sensors [11] are sensitive to illumination levels and sunlight.

Radar sensors have become an emerging technology for mitigating the problems of the abovementioned sensors [12]. Because the radar sensor transmits and receives electromagnetic waves, it can be used in the dark and offers good penetrability for identifying targets obscured by the field of view. Furthermore, it is efficient because a single radar can simultaneously detect multiple targets based on the distance or direction of arrival (DOA) estimation [13]. Hence, various studies have been conducted to detect passengers using radar. A system based on electromagnetic coupling between a transmitter and a receiver patch antenna was developed for passenger occupancy detection [1]. The main limitation of the system is that it can only detect one seat at a time, in addition to a high false alarm rate. A study using a Doppler radar was conducted to detect the occupancy of a seat by thresholding the power of the received radar signal [3]. However, it was tested using an IEE live dummy, not a human. A similar approach was investigated using human subjects; however, experimental results for various states of the passengers were not provided [14]. The presence of passengers for each seat was investigated using IR-UWB radar [15,16]. In addition, using a multiple-input and multiple-output frequency-modulated continuous-wave (FMCW) radar, machine learning algorithms were integrated with DOA estimation to identify occupied seats [17]. In addition, a small child or infant was detected by consistent micro-Doppler effects on the breathing cycles [18]. Although these results were based on certain criteria for the performance evaluation, the criteria were neither rigorous nor clearly defined. Multi-passenger occupancy detection has been studied using the multi-channel FMCW radar [17,18]. However, the multi-channel FMCW radar is difficult to implement and its size and cost is relatively unfavourable compared with the single-channel version.

Herein, based on a single-channel FMCW radar, we provide the results of multi-passenger occupancy detection in a vehicle where various targets, including humans and objects, exist. Althougth conventional studies analyzed the radar signal in time or frequency domain, the characteristics of the vital signs were analyzed based on a time-frequency spectrum in this study. Unlike other movements, the passenger displacement caused by breathing and heartbeat exhibited a periodic intensity change. First, clutter removal [12] was applied to remove the static targets. To quantify a person’s physiological characteristics, we examined various features such as the spectral power and Wiener entropy [19], which quantify the flatness of the power spectral density (PSD). Subsequently, the feature distribution based on the presence or absence of a person in seat 1 (the rear of the passenger seat) and seat 2 (the rear of the driver’s seat) was expressed. We used the classification accuracy as an indicator to evaluate the objective performance. Furthermore, we used linear discriminant analysis (LDA) and maximum likelihood estimation (MLE) as classifiers [20]. To evaluate the classification performance, holdout cross-validation was performed 100 times.

## 2. Materials and Methods

### 2.1. Radar Model

In this study, we used the single-channel FMCW radar. The FMCW radar transmits a linearly modulated radio frequency (RF) signal and processes the received RF signal. After mixing the transmitted and received signals and applying the low-pass filter, the instantaneous frequency (IF) signal x(t,n) is calculated. x(t,n) is composed of the beat frequency $f_{r}$, magnitude M(t,r), and phase P(t,r), as follows [21,22]:(1)$$x\left(t,n\right)=\sum_{r}M\left(t,r\right)cos\left(2\pi f_{r}n+P\left(t,r\right)\right)$$
(2)$$f_{r}=\frac{2Sr}{cF_{s}},$$
where *t* is the scan index, *n* the sample index of a chirp, *r* the range, *S* the slope of the chirp, *c* the speed of light, and $F_{s}$ the sampling frequency.

In Equation (Equation 1), $f_{r}$ is the distance from the radar to the object, M(t,r) the signal power reflected by the object located at *r*, and P(t,r) the time delay of the signal reflected from *r*. We focus on the magnitude M(t,r) for detecting multi-passengers. Using the discrete Fourier transform, $M(t,r_{k})$ is calculated based on x(t,n), as follows:(3)$$X\left(t,r_{k}\right)=\sum_{n=0}^{N-1}x\left(t,n\right)e^{-j\frac{2\pi kn}{N}}$$
(4)$$M\left(t,r_{k}\right)=2\left|X\left(t,r_{k}\right)\right|,$$
where $r_{k}=\frac{c}{2BW}k$ for k=0,...,N−1, BW is the bandwidth of the transmitted signal, and *N* is the total number of samples in a single chirp. $M(t,r_{k})$ can be expressed as shown in Equation (Equation 5) using the radar equation [23].
(5)$$M\left(t,r_{k}\right)=\frac{M_{0}}{4\pi r_{k}^{2}},$$
where $M_{0}$ is the transmitted signal power. Suppose that the object in the range $r_{k}$ vibrates with a small displacement Δ(t) as
(6)$$r\left(t\right)=r_{k}+\Delta \left(t\right)$$ By applying r(t) in Equation (Equation 5) instead of $r_{k}$ and assuming Δ(t) is much smaller than $r_{k}$ ($\left|\Delta \left(t\right)\right|\ll r_{k}$), $M(t,r_{k})$ is approximated based on the Taylor series expansion [24] as follows:(7)$$M\left(t,r_{k}\right)=\frac{M_{0}}{4\pi {\left(r_{k}+\Delta \left(t\right)\right)}^{2}}\approx \frac{M_{0}}{4\pi r_{k}^{2}}\left(1-2\frac{\Delta (t)}{r_{k}}\right)$$

As the small displacement Δ(t) is reflected in the magnitude $M(t,r_{k})$ in Equation (Equation 7), the magnitude $M(t,r_{k})$ is proportional to the small displacement Δ(t). Hence, we can measure a person’s physiological movements based on breathing or heartbeat using $M(t,r_{k})$.

### 2.2. Characteristics of Radar Signals by Breathing

The single-channel FMCW radar used in the study was a BTS60 (bitsensing lnc., Seoul, Korea), and the specifications are shown in Table 1. To understand a person’s physiological characteristics based on radar signals, the time-frequency spectrum of the received radar signal was analyzed. To extract only the frequency characteristics of a person’s physiological movement, we investigated the frequency characteristics using the signal acquired in the presence or absence of a person. Figure 1 shows the time-frequency representation of the received signal after clutter removal in the presence or absence of a person. In Figure 1, the *Y* axis in the first column means the distance after transforming the frequency into the range using Equation (Equation 2). Also, the second column shows the magnitude variation over time at 0.95 m and the third column is the discrete Fourier transform of the signal in the second column. When a person was absent, the magnitude in Figure 1a did not change significantly, and a flat and low spectral magnitude was observed. However, when a person was present, as illustrated in Figure 1b, the magnitude varied over time in the time-frequency spectrum between 0.8 to 1.2 m. At 0.95 m, periodic magnitude variation due to breathing was observed. The frequency analysis shows that the periodic magnitude variation included respiratory (peak at 0.2 Hz) and heartbeat information (peak at 1.3 Hz). The ground-truth respiratory and heart rate are measured using the respiration and electrocardiogram (ECG) sensors and are marked with red circles. Note that the purpose of this paper is not extracting the vital signs but rather detecting human occupancy. The third column in Figure 1 shows that the human vital information is contained in the received radar signal, and this information can be used for occupancy detection. Therefore, the magnitude variation in the time-frequency spectrum was caused by a person’s physiological movement. If we quantify a person’s vital signs based on these physiological movements, then multi-passenger can be distinguished from other stationary objects.

### 2.3. Multi-Passenger Occupancy Detection Inside Vehicle

In Figure 2, the proposed signal processing chain of the multi-passenger occupancy detection algorithm is illustrated. When detecting multi-passenger in a vehicle, the vital sigsares a useful indicator and must be measured quantitatively. Prior to quantifying a person’s physiological movement, clutter removal was applied to remove the static targets from the received radar signal. By removing the complex average across all scans, information regarding the stationary object was removed, and the signals returned from moving objects remained. Clutter removal, as expressed below, was performed:(8)$$X_{CR}\left(t,r_{k}\right)=X\left(t,r_{k}\right)-\frac{1}{T}\sum_{\tau =t-T+1}^{t}X\left(\tau ,r_{k}\right),$$
where *T* is the window size. When *T* is too short, it is difficult to quantify a person’s vital signs because the number of data samples for the frequency analysis is too small. On the other hand, when *T* is too long, a long time delay arises in passenger occupancy detection.

After clutter removal, we used two features, namely the power and Wiener entropy, to quantify the spectral characteristics of the received radar signals. Power was selected because it is a widely used and simple measure of spectral characteristics, whereas the Wiener entropy can measure the spectral inequality. The power of the signal is measured at each time instant, as follows:(9)$$pow\left(t,r_{k}\right)=\sum_{\tau =t-T+1}^{t}{\left|X_{CR}\left(t,r_{k}\right)\right|}^{2}$$ The spectral power was low when a person was absent. However, when a person was present, the breath and heartbeat of the person resulted in a significant variation in the spectrum, and hence a significant spectral power.

The Wiener entropy can measure the flatness or inequality of PSD. Using these characteristics, the Wiener entropy quanitified the periodicity of PSD. The Wiener entropy is calculated as follows:(10)$$PSD_{t}\left(f_{l},r_{k}\right)=\sum_{\tau =t-T+1}^{t}X_{CR}\left(\tau ,r_{k}\right)e^{-j\frac{2\pi l\tau }{T}}$$
(11)$$Wienerentropy\left(t,r_{k}\right)=\frac{\sqrt[{T}]{\prod_{l=1}^{T}\left|PSD_{t}\left(f_{l},r_{k}\right)\right|}}{\frac{1}{T}\sum_{l=1}^{T}\left|PSD_{t}\left(f_{l},r_{k}\right)\right|}$$ When a person is absent as in Figure 1a, the PSD is constant or flat, so the Wiener entropy becomes high. Meanwhile, when a person is present, as in Figure 1b, PSD is concentrated in the respiratory frequency band, so the Wiener entropy becomes low. Hence, the Wiener entropy quanitifies the person’s physiological characteristics and can be used as a feature for multi-passenger occupancy detection.

Figure 3 shows the results of the two proposed features applied to the received radar signal acquired inside the vehicle. Passengers A and B boarded at 30 and 60 s, respectively. In addition, passengers B and A entered at 240 and 270 s, respectively. Figure 3a,b show the results of the power and Wiener entropy, respectively. As shown in Figure 3a, the power changed depending on whether a person was present. The results shown in Figure 3b indicate that the Wiener entropy can reflect the occupancy of passengers.

## 3. Results

To validate the proposed features for multi-passenger occupancy detection, experiments were conducted inside a vehicle using the single-channel FMCW radar. The proposed two features were compared, and multi-passenger occupancy detection performance was evaluated in terms of accuracy. The vehicle used was the Grandeur by Hyundai Motors, and the radar was installed on a B-pillar behind the passenger rear seat. The experiment was conducted on two rear seats. The window length *T* sets to four seconds, which corresponds to 80 scans.

Figure 4 shows the results of the two proposed features applied to the radar signal acquired inside the vehicle under various conditions. Figure 4a shows the case in which five passengers entered the vehicle. The spectral features changed based on the time at which a person entered the vehicle. In addition, Figure 4b shows an experiment involving static objects such as a bag and a bookshelf. As shown in Figure 4b, the person appeared at time 30–60 and 150–210 s, whereas the objects appeared at time 30–90, 120–180 and 240–270 s. When a person was present, the features in Figure 4b changed depending on the person’s presence. However, for the case involving static objects, the features were similar to those in the case where a person was absent. This implies that the static objects were not wrongly detected as passengers.

The distance between the radar and the rear left seat (seat 1) was 0.8 to 1.0 m, whereas that between the radar and the rear right seat (seat 2) was 1.1 to 1.3 m. Based on this information, the spectral features were averaged over the corresponding range bins. Figure 5 shows the average mean of the proposed features at seats 1 and 2 under various rear-occupancy conditions. In each figure, the first row shows images from the experiments, whereas the second and third rows show the results of the proposed features, i.e., power and the Wiener entropy. In addition, the second and third columns show the average mean of the feature over the range bins corresponding to seats 1 and 2, respectively. Figure 5a shows the case in which five passengers entered the vehicle. In addition, Figure 5b shows a case in which one person and static objects were placed on the rear seat. Although the static objects were present, the average mean of the proposed features was similar to that of the case where a person was absent. The average means of both the power and Wiener entropy changed significantly and facilitated multi-passenger occupancy detection.

To characterize the distribution of the passenger rear occupancy, first, the rear occupancy cases were categorized into four cases: (N) no passenger, (S1) passenger in seat 1, (S2) passenger in seat 2, and (S12) passengers in both seats 1 and 2. The feature distributions are shown in Figure 6a,b. Subsequently, using the Gaussian model, each case was modeled in a two-dimensional (2D) feature space, as shown in Figure 6c,d. The power distributions for the four cases are sufficiently distinguished in Figure 6a,c. Although the Wiener entropy shown in Figure 6b,d had a limit, as it is more spread than power, the 2D Gaussian model of the Wiener entropy was able to classify the four cases.

In this study, LDA and MLE were used as classifiers. LDA was selected because it is a simple and widely used method, and MLE is known as a statistically optimum method. Figure 7 shows the distribution of each seat’s feature and the decision boundary of the LDA and MLE.

To evaluate the classification performance, we performed holdout cross-validation 100 times, and the dataset was randomly segregated into training and test data at a ratio of 8:2, respectively. For training and classification, the average means of both the power and Wiener entropy at seats 1 and 2 were considered. Figure 8 shows the classification results of the multi-passenger occupancy detection using LDA and MLE. The classification accuracy was calculated and averaged over 100 independent trials. In the LDA, confusion occurred between the case involving one passenger in the rear right seat and the case where the passengers were in both seats. However, using MLE, this false alarm decreased significantly. The average accuracy of the LDA and MLE was 96.14 % and 98.88 %, respectively. The accuracy of multi-passenger occupancy detection was higher when MLE was employed, althought the complexity increased.

## 4. Discussion

For multi-passenger occupancy detection inside the vehicle, the power and the Wiener entropy were introduced as features, and the classification was performed using LDA and MLE. Instead of using conventional multi-channel FMCW radar, multi-passenger occupancy detection using single-channel FMCW radar was conducted, and achieved the average accuracy of 96.14 % by LDA and 98.88 % by MLE. This paper has performed experiments with passengers and static objects. However, the various vibrating objects, such as a water bottle and a mobile phone, can cause an unexpected false alarm. Therefore, the future work needs to improve the stability of the multi-passenger occupancy detection by reflecting the characteristics of various dynamic objects.

## 5. Conclusions

In this study, we analyzed a person’s physiological characteristics in the time-frequency spectrum based on the single-channel FMCW radar and proposed features using these characteristics to perform multi-passenger occupancy detection inside a vehicle. When a person was present, the breath and heartbeat of that person resulted in a significant variation in the spectrum. Consequently, the spectral power increased and was concentrated in a narrow frequency band. Hence, the passengers were detected by quantifying the spectral characteristics. Power and Wiener entropy were proposed as features to quantify physiological movements.

To evaluate the performance of multi-passenger occupancy detection, the feature distribution based on the presence or absence of a person in seats 1 and 2 was obtained. Classification was performed using the average means of both the power and Wiener entropy at seats 1 and 2. The performance of multi-passenger occupancy detection was evaluated using the LDA and MLE. The results indicated that the proposed features effectively detected multiple passengers in a vehicle using a single-channel FMCW radar.

## Figures and Tables

**Figure 1 entropy-23-01472-f001:**
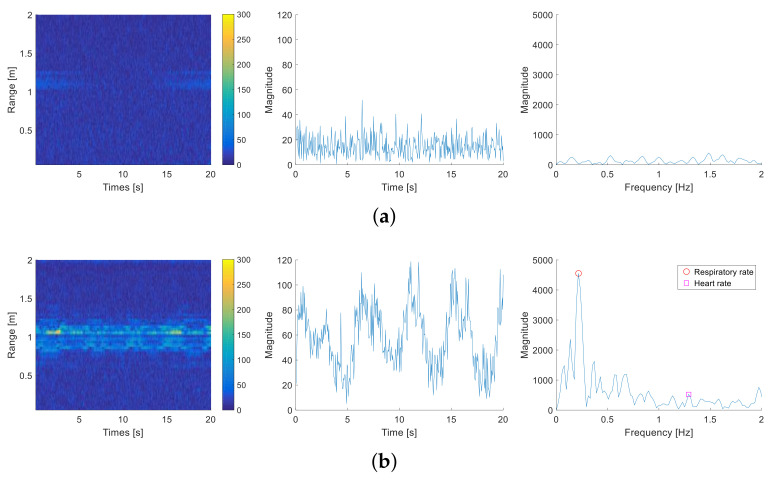
Time-frequency representation of received radar signal, and analysis of magnitude variation at 0.95 m in time and frequency domains when a person is (**a**) absent and (**b**) present.

**Figure 2 entropy-23-01472-f002:**
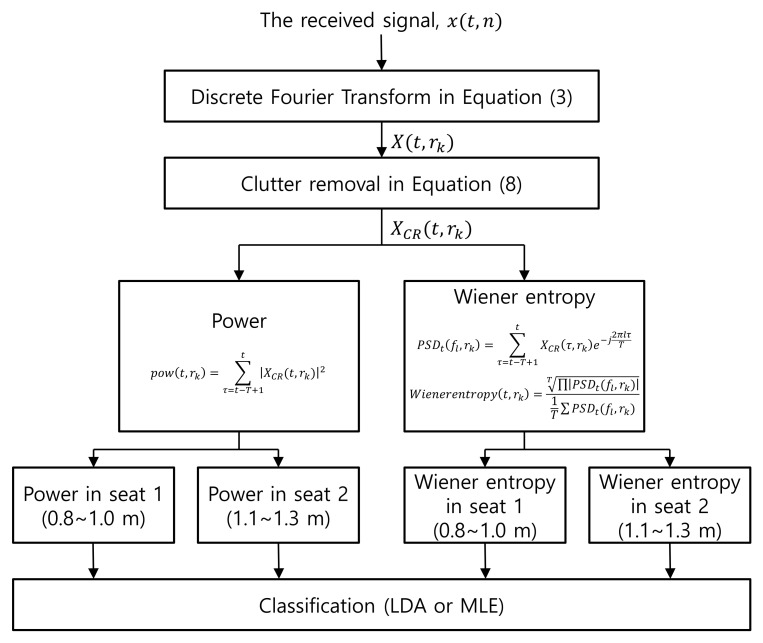
The proposed multi-passenger occupancy detection algorithm.

**Figure 3 entropy-23-01472-f003:**
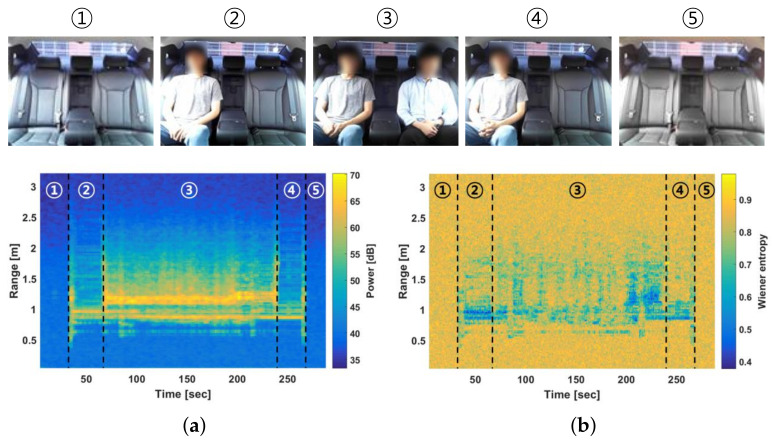
Two spectral features based on various passengers’ presence inside vehicle. (**a**) Spectral power; (**b**) Wiener entropy.

**Figure 4 entropy-23-01472-f004:**
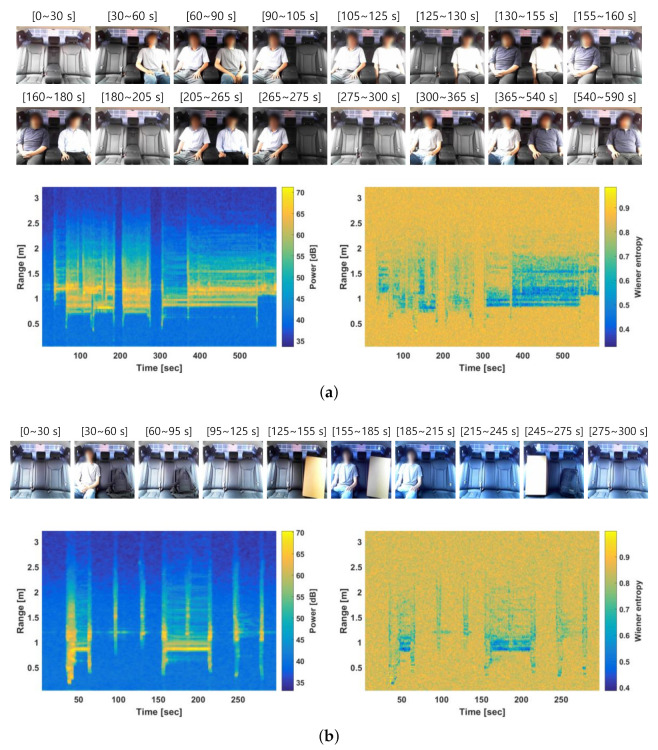
Results of two spectral features inside a vehicle under various conditions. (**a**) Five passengers boarding and alighting vehicle. (**b**) One person and static objects placed on rear seat.

**Figure 5 entropy-23-01472-f005:**
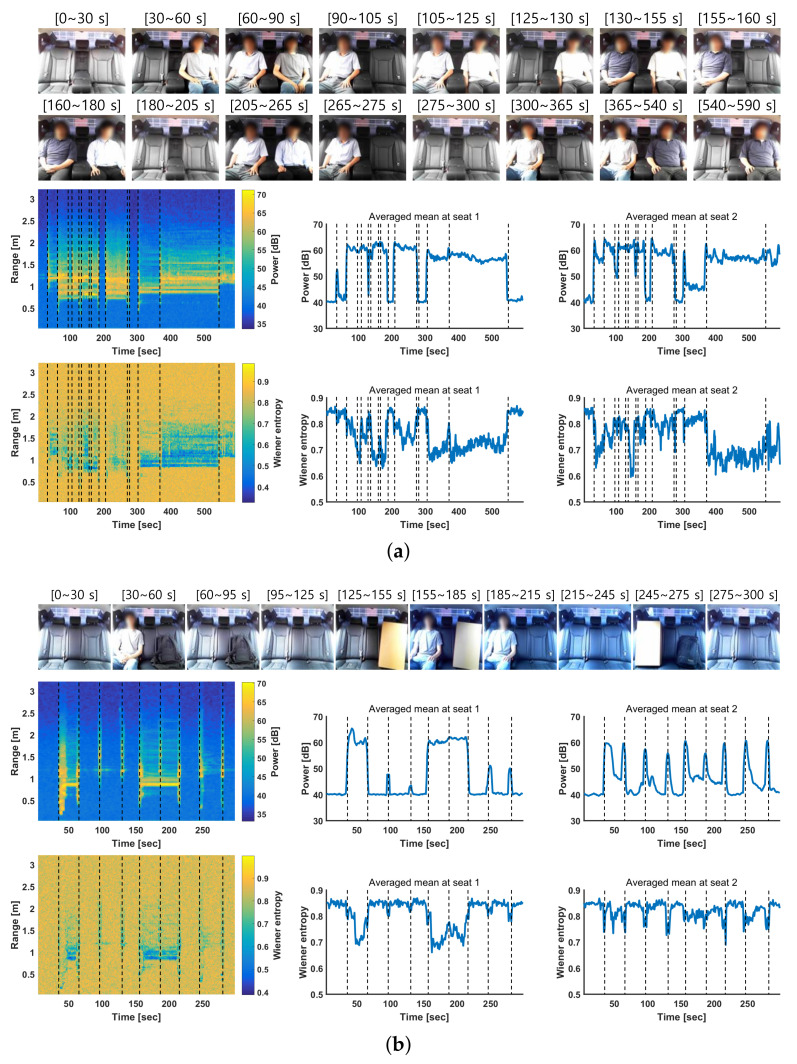
Mean of features at seat 1 (0.8 to 1 m) and seat 2 (1.1 to 1.3 m) under various rear occupancy conditions. (**a**) Five passengers entering a vehicle. (**b**) One person and static objects placed on the rear seat.

**Figure 6 entropy-23-01472-f006:**
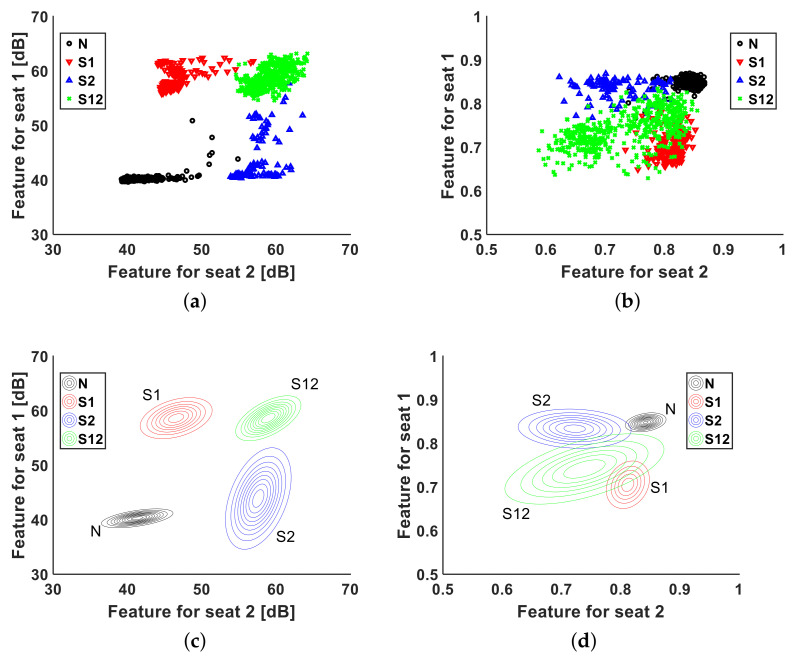
Feature distributions and 2D Gaussian models from four rear occupancy cases: (N) no passenger, (S1) passenger in seat 1, (S2) passenger in seat 2, and (S12) passengers in both seats 1 and 2. (**a**) Feature distribution of power. (**b**) Feature distribution of Wiener entropy. (**c**) 2D Gaussian model of power. (**d**) 2D Gaussian model of Wiener entropy.

**Figure 7 entropy-23-01472-f007:**
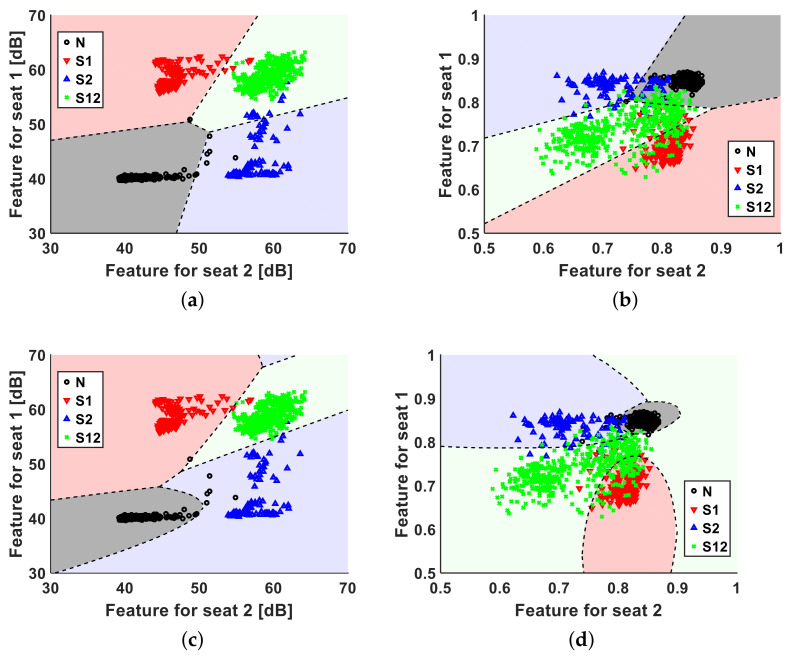
Distributions for each seat’s feature, and the decision boundary of LDA and MLE under four cases. (**a**) Decision boundary of power based on LDA. (**b**) Decision boundary of Wiener entropy based on LDA. (**c**) Decision boundary of power based on MLE. (**d**) Decision boundary of Wiener entropy based on LDA.

**Figure 8 entropy-23-01472-f008:**
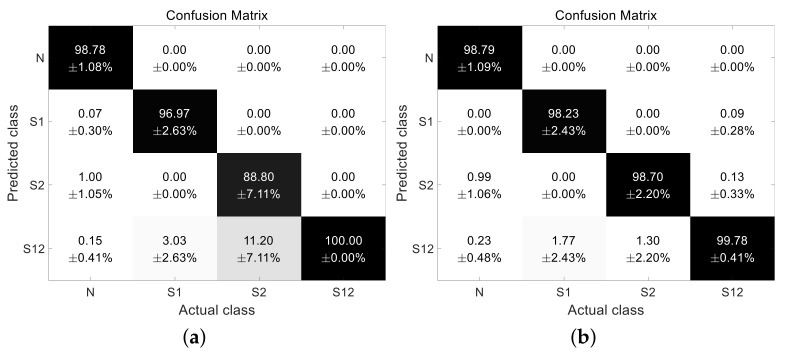
Classification result of passenger detection. (**a**) Confusion matrix based on LDA. (**b**) Confusion matrix based on MLE.

**Table 1 entropy-23-01472-t001:** Single-channel FMCW radar specification.

Parameter	Value
Center frequency	61 GHz
Bandwidth	6 GHz
Chirp duration	128 μs
Sampling frequency	2 MHz
Scan interval	50 ms
Tx antenna	1 channel
Rx antenna	1 channel
Beamwidth (Azimuth)	$\pm 65^{\circ }$
Beamwidth (Elevation)	$\pm 65^{\circ }$

## Data Availability

Not applicable.

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
