# Peer review of "Single-Channel FMCW-Radar-Based Multi-Passenger Occupancy Detection Inside Vehicle"

_entropy, 2021, doi:10.3390/e23111472_

Round 1
Reviewer 1 Report
In this manuscript, the authors proposed a scheme to detect multiple passengers inside a vehicle. The proposed scheme is based on single-channel frequency-modulated continuous-wave (FMCW) radar. The characteristics of the vital signs were analyzed based on a time-frequency spectrum. Following that, clutter removal is applied to remove the static targets. The spectral power and Wiener entropy are proposed as features to quantify physiological. In addition, the authors also evaluated the passenger occupancy detection performance using linear discriminant analysis and maximum likelihood estimation. Lastly, numerical results show that the proposed scheme is effective in passenger occupancy detection. Overall, the paper is quite well written with good presentation. The idea presented in this manuscript is interested and it has been demonstrated to perform reasonably well as compared with benchmark. The proposed scheme might be useful for system designer. I recommend the paper to be accepted for publication in this journal.Author Response
Thank you for your positive feedback.
Reviewer 2 Report
General Comments:
The paper deals with passenger detection in vehicles and presents an approach for multi-passenger detection inside a vehicle using a single-channel frequency-modulated continuous-wave radar sensor. The radar signal is analyzed using time-frequency analysis, where features are extracted for passenger detection. The spectral power and Wiener entropy are considered as features linked to breathing and heartbeat. The topic is important for transport safety, and the proposed approach is interesting.
Specific Comments:
1. Sub-Section 2.2:
a) It is mentioned that time-frequency analysis has been used in this study, however, Equation (3) is using just Fourier Transform. Probably time-range is meant instead, as Figures 1 and 2 suggest. Figure 1 uses time and frequency separately.
b) On Page 3, please clarify what bandwidth means.
c) It is unclear how to specify the heart-rate and the respiratory-rate in Figure 1, as there are multiple peaks in the spectrum. In fact, the interaction between the radar LFM signal and the human body has not been analyzed.
2. Section 2.3:
a) Equations (8)-(11) depend on the window length T. Please clarify the effect of this window length on the performance of the proposed system.
b) The behavior of Weiner entropy for presence/absence detection should be clarified. Are there no other measures?
3. There should be a quantitative comparison with some other methods based on complexity-accuracy tradeoff.
Author Response
Thank you for valuable comments, and an opportunity to address the comments.
Please see the attachment.

Reviewer 3 Report
- The word Multi-Passenger has been used in the title and in the first line of abstract. However, it is strange that this word (Multi-Passenger) has not been used in the remaining article. Similarly, the word single-channel is only used in abstract, introduction and conclusion. This characteristic has not been discussed in the remaining article.
- The proposal is based on a single-channel FMCW radar. The motivation behind this choice is not presented in introduction. In other words, the research gap has not been properly identified.
- How the proposed method is different from existing implementations ? Novelty should be described explicitly.
- Another major limitation of introduction is the lack of significance. The achieved outcomes with the proposed method as well as their significance are not described. Without this important information, the contributions of the article towards the body of knowledge can not be judged.
- Section 2 (materials and methods) is supposed to provide the design details of the proposed system. However, the article under consideration is mixing the design details with implementation details.
- The concepts have been illustrated mathematically, however, the readability of this article must be improved through some illustrative design representations (structure as well as behavior).
- The achieved results have not been compared with state-of-the-art implementations.
- Performance comparison should also include the reason behind the increase/decrease of performance
- Performance must be compared in terms of somee well defined attributes.
- There should be a comprehensive discussion section which can describe the significance of achieved results.
- The discussion section should also describe the limitations of the proposed method ??
Author Response

(The authors gave the same response as above.)

Round 2
Reviewer 2 Report
The Authors have addressed the Reviewer’s comments in detail. The current version of the paper represents a useful application of entropy to solve real-life problems. I find it suitable for publication in MDPI Entropy.
Reviewer 3 Report
The raised comments have been addressed.
The article can be published in its current form.